# Identification of Drought Stress-Responsive Genes in Rice by Random Walk with Multi-Restart Probability on MultiPlex Biological Networks

**DOI:** 10.3390/ijms25179216

**Published:** 2024-08-25

**Authors:** Jiacheng Liu, Liu Zhu, Dan Cao, Xinghui Zhu, Hongyan Zhang, Yinqiong Zhang, Jing Liu

**Affiliations:** 1College of Information and Intelligence, Hunan Agricultural University, Changsha 410128, China; liujc@stu.hunau.edu.cn (J.L.); zhuliu@stu.hunau.edu.cn (L.Z.); zhuxh@hunau.edu.cn (X.Z.); lemonhunau@hunau.edu.cn (Y.Z.); lj1174298794@stu.hunau.edu.cn (J.L.); 2College of Science, Central South University of Forestry and Technology, Changsha 410004, China; caodan@csuft.edu.cn

**Keywords:** rice, drought stress-responsive genes, MultiPlex biological networks, random walk with multi-restart probability, eigenvector centrality

## Abstract

Exploring drought stress-responsive genes in rice is essential for breeding drought-resistant varieties. Rice drought resistance is controlled by multiple genes, and mining drought stress-responsive genes solely based on single omics data lacks stability and accuracy. Multi-omics correlation analysis and biological molecular network analysis provide robust solutions. This study proposed a random walk with a multi-restart probability (RWMRP) algorithm, based on the Restarted Random Walk (RWR) algorithm, to operate on rice MultiPlex biological networks. It explores the interactions between biological molecules across various levels and ranks potential genes. RWMRP uses eigenvector centrality to evaluate node importance in the network and adjusts the restart probabilities accordingly, diverging from the uniform restart probability employed in RWR. In the random walk process, it can be better to consider the global relationships in the network. Firstly, we constructed a MultiPlex biological network by integrating the rice protein–protein interaction, gene pathway, and gene co-expression network. Then, we employed RWMRP to predict the potential genes associated with rice tolerance to drought stress. Enrichment and correlation analyses resulted in the identification of 12 drought-related genes. We further conducted quantitative real-time polymerase chain reaction (qRT-PCR) analysis on these 12 genes, ultimately identifying 10 genes responsive to drought stress.

## 1. Introduction

Abiotic stresses, particularly drought stress, exert significant impacts on both the yield and quality of rice. The exploration of genes in rice responsive to drought is imperative for the development of drought-resistant varieties. The regulatory network underlying rice drought tolerance entails multiple genes, and the identification of genes associated with drought stress, through the analysis of rice omics data, is pivotal for enhancing rice survival under drought conditions. It is crucial to recognize that distinct omics disciplines provide insights into gene transcription, expression, translation, and modifications within an organism from diverse perspectives. Sole reliance on a single omics dataset for analysis may lead to an incomplete understanding of the organism’s expression profile. Therefore, the adoption of multi-omics correlation analysis and molecular biology network analysis is paramount. This approach not only enables a comprehensive exploration of drought-resistant genes but also facilitates a deeper comprehension of the intricate molecular mechanisms involved [1].

Traditionally, large-scale biological networks exhibit characteristics of small-world and scale-free networks [2]. Genes with similar functions and features typically demonstrate modularity within biological networks. Various methods have been employed to explore key genes in biological networks and investigate interaction patterns among genes, thereby revealing specific regulatory mechanisms, such as Gene Co-expression Network Analysis [3] and Genetic Association Analysis [4]. Among the commonly utilized methods for this purpose are Weighted Gene Co-expression Network Analysis $\left(WGCNA\right)$ [5] and Genome-Wide Association Study $\left(GWAS\right)$ [6]. WGCNA groups genes exhibiting similar expression patterns and examines the correlation between modules and particular traits to find central genes within each module. GWAS comprehensively explores variation patterns in the genome and identifies genetic variations associated with specific phenotypes. Key genes are uncovered by analyzing the correlation between genetic and phenotypic variations. However, it is important to note that in situations with limited sample data, co-expression modules obtained from WGCNA may exhibit instability. Additionally, GWAS, often involving a large number of Single Nucleotide Polymorphisms $\left(SNPs\right)$, can lead to a higher false-positive rate.

The Restarted Random Walk $\left(RWR\right)$ algorithm [7] is a probability-based mathematical model designed to explore and identify important nodes or paths within complex networks. In recent years, the RWR algorithm and its derived innovations have found widespread application in various computational biology problems [8]. Zhu et al. [9] constructed a dual-layer network utilizing protein–protein interaction networks and gene co-expression networks. They applied the RWR algorithm on this dual-layer network to predict the rice drought stress-responsive genes. Li Y et al. [10] introduced an RWR-H algorithm based on heterogeneous networks for predicting risk-causing genes. The heterogeneous network consists of a PPI network, a disease phenotype network, and a protein–disease phenotype association bipartite graph. This algorithm identified 18 gene–disease associations, which are mostly supported by literature evidence. Valdeolivas A et al. [11] proposed the RWR-MH algorithm based on RWR-H, extending the single protein interaction network to multiple biological networks. This led to the construction of a single gene multiple relationship network, significantly improving the predictive performance of target genes. The algorithm was successfully applied to predict candidate genes for Wiedemann-Rautenstrauch syndrome. These algorithms have innovatively approached network construction, enhancing their performance in the random walk processes. They assign the same restart probability to all nodes in the network, constraining the exploration of target genes. This study introduces the innovative Random Walk with Multi-Restart Probability $\left(RWMRP\right)$ algorithm. By evaluating node importance in the network through eigenvector centrality [12] and assigning diverse restart probabilities to nodes based on their respective importance, this strategy facilitates a comprehensive exploration of the interconnectivity between all nodes and seed nodes from a global perspective.

This study constructed protein interaction networks, gene co-expression networks, and gene pathway networks based on rice multi-omics data, with each node corresponding to genes or proteins. These three networks were then merged into a MultiPlex network, allowing multiple edges between pairs of nodes. Subsequently, known drought stress genes from the China Rice Data Center were employed as seed nodes in the MultiPlex network. The RWMRP algorithm was utilized to predict potential drought stress genes in rice. Enrichment and association analyses were conducted on these potential genes to further identify drought stress candidates. Finally, qRT-PCR analysis [13] was performed on the candidate genes, leading to the identification of 10 drought stress-responsive genes.

## 2. Results

### 2.1. Performance Prediction Analysis of the RWR and the RWMRP on Various Biological Molecular Networks

#### 2.1.1. Performance Prediction Analysis of the RWR on Various Biological Molecular Networks

The study constructed single-layer, double-layer, and multi-layer networks based on the multi-source protein–protein interaction network MetaPPI, gene co-expression network CoEx, and gene pathway network Pathway, respectively. The predictive performance of the Restarted Random Walk algorithm on these networks was compared using the LOOCV strategy. The performance metric utilized in this evaluation was the ratio of left-out genes with rankings equal to or less than 300 to the total number of genes in the test set. The single-layer network comprised MetaPPI, CoEx, and Pathway networks, separately. The double-layer networks were formed by pairwise fusion of these networks (MetaPPI_CoEx, MetaPPI_Pathway, CoEx_Pathway). The multi-layer network, named MetaPPI_CoEx_Pathway, resulted from merging the three individual networks. As illustrated in Figure 1, the overall predictive performance increased with the number of network layers. The fusion of multiple omics data supplemented the performance of single omics data, leading to significantly improved predictive outcomes, with approximately 66% of the left-out genes located within the top 300 genes.

#### 2.1.2. Performance Prediction Analysis of the RWMRP on Various Biological Molecular Networks

This section evaluated the performance of the Multiple Probability Restart Random Walk (RWMRP) algorithm on the multi-layer networks using the same data as the previous section. As illustrated in Figure 2, the predictive performance curves of RWMRP on different networks closely mirrored those of RWR on the same networks. With an escalation in the number of network layers, the predictive performance also exhibited improvement, reaching approximately 68.6% of left-out genes located within the top 300 genes.

#### 2.1.3. The Multiple Biological Molecular Networks Exhibit Superior Performance

As depicted in Figure 1 and Figure 2, the predictive performance demonstrates an upward trend with the augmentation of network layers. Notably, the three-layer network exhibits superior overall prediction performance compared to other networks. This observation suggests that the integration of multi-omics data significantly enhances the predictive efficacy compared to single-omics data.

#### 2.1.4. Enhanced Predictive Performance of RWMRP Compared to RWR

As shown in Figure 1 and Figure 2, the proportions of missing genes among the top 300 predicted genes by RWR and RWMRP on the MetaPPI_CoEx_Pathway network are 66% and 68.6%, respectively. In comparison to the traditional restart random wandering algorithm, the RWMRP algorithm, based on eigenvector centrality, exhibits improved predictive capabilities for abiotic adversity-responsive genes within the multiple biological molecular networks constructed in this chapter. This indicates that leveraging eigenvector centrality to assess the importance of nodes in the network effectively enhances prediction performance.

### 2.2. Obtaining Potential Drought Stress-Responsive Genes Based on RWMRP

Drought stands as one of the most prevalent adverse environmental challenges confronting rice cultivation. Consequently, the accurate prediction of drought-resistant genes holds paramount importance for augmenting both the yield and quality of rice. In this study, we applied the RWMRP algorithm to the MetaPPI_CoEx_Pathway network in rice. We utilized known drought-tolerant genes downloaded from the China National Rice Data Center as seed nodes in the biological network. Subsequently, genes predicted by RWMRP within the top 300 rankings were identified as potential genes associated with drought stress.

### 2.3. Enrichment Analysis of Potential Genes for Drought Stress

This study conducted gene function enrichment analysis on potential genes related to drought stress predicted by RWMRP using the online enrichment analysis tool AmiGO (http://amigo.geneontology.org (accessed on 25 April 2024)) [14]. The enrichment results indicated that the potential genes were significantly enriched in multiple relevant gene ontology (GO) terms across three major categories: biological process (BP), molecular function (MF), and cellular component (CC). As presented in Table 1, a subset of GO terms with a *p*-value below 0.01 was selected for presentation and documentation. The enrichment analysis revealed that the potential genes predicted for drought stress in rice exhibited significant enrichment in response to abiotic stress, temperature stimuli, and environmental stimuli within the biological processes category. Regarding molecular functions, these potential genes were notably enriched in protein serine/threonine kinase activity, lycopene beta-cyclase activity, galactinol synthase activity, and ascorbate peroxidase activity. These findings validate the effectiveness of the methodology employed in this study for predicting potential genes associated with drought stress in rice.

### 2.4. Obtaining Candidate Genes for Drought Stress Based on Association Analysis

Our candidate genes were selected from the subset of $GP_{j}$ genes demonstrating three or more interactions with known drought genes $\left(KG_{i}\right)$, each with an $S=\left(PG_{j},KG_{i}\right)$ score ≥ 0.9. Utilizing Cytoscape software (Version 3.9.0), we visualized the relationships between drought stress potential genes and known genes, illustrated in Figure 3.

Ultimately, 12 genes linked to drought stress were definitively identified. The details of the candidate genes retrieved from the STRING database are provided in Table 2.

In response to drought stress, plants employ various stress tolerance mechanisms including omics-level responses, antioxidative defense mechanisms, osmotic protection, and osmotic regulation [15]. In the drought resistance mechanism of rice, zeta-carotene desaturase [16], lycopene [17], trehalose-6-phosphate synthase/phosphatase [15], monodehydroascorbate reductase [18], serine/threonine protein kinases [19], and other hormones play crucial roles. This further validates the effectiveness of our predicted candidate genes.

### 2.5. The Results of qRT-PCR

We used qRT-PCR to measure the expression of 12 candidate genes under drought conditions. The results are shown in Figure 4; after 4 h of drought treatment, the expression levels of 10 out of 12 candidate genes (Os07g0204900, Os09g0397300, Os02g0196000, Os04g0432000, Os09g0516500, Os06g0729000, Os03g0405000, Os02g0707000, Os02g0707100, and Os12g0541300) significantly increased. This suggests these genes play a role in the rice drought stress response. Details of the primer design for reference genes and candidate genes are shown in Appendix A, and the original data of qRT-PCR experiment are shown in Appendix A, which contains the Cycle Threshold (CT) values and relative expression levels of all genes.

## 3. Discussion

The tolerance of rice to drought stress is generally regulated by multiple genes. When rice is subjected to drought stress, the expression levels of stress-responsive genes undergo changes, releasing stress signaling molecules to modulate its drought tolerance [20]. In this study, we applied the RWMRP algorithm to the rice multi-biological molecular network to predict drought stress genes. Experimental validation confirmed that the multi-layered network can compensate for the limitations of single-omics data. Moreover, assigning different restart probabilities to nodes based on eigenvector centrality significantly enhanced the algorithm’s predictive performance.

As shown in Figure 4, the qRT-PCR results indicate that among the candidate drought stress genes predicted in this study, a total of 10 genes responded to drought stress. Among these, the expression levels of eight genes were significantly upregulated after 4 h of PEG treatment. According to the experimental results of other researchers, seven of these genes (Os04g0432000 [21], Os06g0729000 [17], Os02g0190600, Os07g0204900 [22], Os09g0397300 [23], Os02g0707000, and Os02g0707100 [18]) are involved in the drought stress-response process in rice, which is consistent with the qRT-PCR results of this study. Additionally, orthologs of these genes in other organisms also play significant roles in drought resistance.

Under adverse conditions, protein phosphorylation and dephosphorylation constitute essential reactions in energy metabolism and signal transduction. Protein kinases, such as SnRK2s, catalyze protein phosphorylation [19]. Members of the SnRK2s kinase family (SAPK1-SAPK10) have been extensively reported to be involved in regulating the response to abiotic stress in rice [24]. Under drought stress, the expression levels of SAPK7 and transcription factor WRKY24 in D. catenatum are upregulated [25]. Dehydration is a major factor causing damage to plants under high osmotic stress. Studies on rice [24], Arabidopsis [26], and tobacco [27] have found that the expression levels of all ten members of SnRK2s vary under high osmotic stress. Additionally, the direct homologs of SAPK7 (Os04g0432000) in wheat exhibit rapid and high expression under various stressors such as drought, cold, salt, and high osmotic stress, suggesting its role as a general stress-responsive gene [28]. The plant antioxidative defense system comprises enzymatic antioxidants (including monodehydroascorbate reductase, peroxidases, and catalases) and non-enzymatic antioxidants (carotenoids, ascorbic acid, glutathione, and β-carotene) [29]. Under stress conditions such as drought, plants produce reactive oxygen species (ROS), and the accumulation of high ROS levels can lead to DNA oxidation and other damages, thereby restricting plant growth and development [30]. Ascorbic acid (AsA) is an important antioxidant that directly neutralizes ROS. Studies on plants such as rice [31], maize [32], and wheat [33] have shown that monodehydroascorbate reductase genes (Os02g0707000, Os02g0707100) catalyze the conversion of monodehydroascorbate to ascorbic acid, enhancing plant resistance to temperature, drought, and other abiotic stresses [18]. Carotenoids are natural pigments with fundamental physiological functions in all photosynthetic organisms [16]. They help plants adapt to abiotic stress by antioxidative action and regulation of abscisic acid (ABA) synthesis. Phytoene synthase (PSY) (Os06g0729000) mediates the first step of carotenoid biosynthesis by catalyzing the isomerization of GGPP (geranylgeranyl pyrophosphate) to 15,15′-cis-phytoene, thereby promoting carotenoid synthesis [17]. Studies have reported that the direct homologs of PSY1 play significant roles in the drought response mechanisms of plants such as tomato [34], apple, and Arabidopsis [35]. The overexpression of lycopene beta-cyclase genes (LCYB) (Os02g0190600, Os07g0204900) has also been shown to enhance drought and salt tolerance in rice [36], Arabidopsis [37], tomato [38], and sweet potato [39]. In osmotic protection mechanisms, trehalose is an important osmoprotectant [15]. In rice, Os09g0397300 belongs to trehalose-6-phosphate synthase (TPP), which is involved in trehalose synthesis. Under stress, the trehalose content in plants rapidly increases, forming a glassy structure that helps stabilize proteins and lipids in membranes under low temperature and dehydration conditions, thus conferring strong dehydration resistance and enhancing plant tolerance to drought stress [23]. AtTPPF [40] and AtTPPI [41] have been experimentally verified as two subtypes of genes in Arabidopsis that are associated with the drought stress response.

In summary, the application of RWMRP across MultiPlex networks has the potential to reveal crucial genes associated with drought stress. This bears significant implications for the exploration of drought stress-responsive genes in plants.

## 4. Materials and Methods

### 4.1. Data Sets and Biological Molecular Network

#### 4.1.1. Protein–Protein Interaction Network

We integrated data from various public rice PPI databases, including STRING [42], PRIN [43], and RicePPINet [44]. This integration enhanced the richness of information within the PPI network while minimizing sample bias and noise. The study extracted 8,949,049, 708,819, and 76,586 pairs of protein interactions from the STRING (Version 11.5) (https://version-11-5.string-db.org/cgi/download?sessionId=b7jGX7Q9jxkk (accessed on 8 October 2023)), RicePPINet (the data can be downloaded from https://netbio.sjtu.edu.cn/riceppinet/ (accessed on 8 October 2023)), and PRIN databases (the data can be downloaded from https://bis.zju.edu.cn/prin/download.do (accessed on 8 October 2023)), respectively. These data were obtained on 20 June 2023. To ensure consistency in the gene nomenclature across various databases containing rice protein–protein interaction pairs, a matching table of 33,675 protein–gene name pairs was obtained from the RAP-DB database (https://rapdb.dna.affrc.go.jp/ (accessed on 8 October 2023)) [45]. Additionally, protein names that did not find a match were removed. Consequently, the remaining protein interaction pairs in the STRING, RicePPINet, and PRIN databases were 8,949,049 pairs, 673,489 pairs, and 55,211 pairs, respectively. Subsequently, based on this data, protein interaction pairs from different sources were merged to construct a multi-source rice protein interaction network, termed MetaPPI. It encompasses 29,848 nodes and 1,967,206 edges.

#### 4.1.2. Gene Co-Expression Network

Zhang et al. [46] retrieved all publicly available RNA-seq transcriptomes of rice from NCBI’s SRA database. After implementing rigorous quality control measures on the samples, they retained 8456 sets of high-quality RNA-seq transcriptome data. The data can be downloaded from https://figshare.com/ndownloader/files/35905865 (accessed on 8 October 2023). In this study, we standardized the gene expression data for these 8456 sets of rice samples extracted from their supplementary files. Subsequently, a rice gene co-expression network $\left(CoEx\right)$ was established based on Pearson correlation coefficients, resulting in a network comprising 10,210 nodes and 458,466 edges.

#### 4.1.3. Gene Pathway Networks

The KEGG [47] database (https://www.genome.jp/kegg/pathway.html (accessed on 8 October 2023)), which provides pathway data for rice along with downloadable links, was utilized in this study. All rice pathway data were systematically retrieved from the KEGG Pathway database, totaling 104 pathways as of 8 January 2023.

To analyze the rice gene pathways obtained from the KEGG database, the R package KEGGgraph (Version 1.64.0) [48] was employed. Following parsing with KEGGgraph, the gene pathway diagrams were transformed into a format resembling protein–protein interaction pairs. In this format, gene associations were inferred from the connections between gene products within the nodes of the pathway diagrams. The resulting gene pathway network comprises 2336 nodes and 26,911 edges.

#### 4.1.4. Multiple Biological Molecular Networks

These three networks were integrated into a trilayer network as illustrated in Figure 5. The multiple networks share the same node set, where black nodes represent the genes/ proteins present in each individual layer, while red nodes represent genes/proteins existing in only one or any two layers. Additionally, to compare the predictive performance of the proposed algorithms on different networks, various two-layer biological molecular networks were formed by pairwise combinations of protein–protein interaction, gene co-expression, and gene pathway networks.

#### 4.1.5. Known Genes for Abiotic Stress-Responsive

This study retrieved 2082 known resistance genes associated with 12 types of abiotic stress from the China Rice Data Center (https://www.ricedata.cn/ (accessed on 25 December 2023)). The subset of known abiotic stress genes was employed in the LOOCV $\left(Leave-One-OutCross-Validation\right)$ [9] strategy to assess the predictive performance of RWMRP and RWR algorithms.

Among the obtained known abiotic genes, 345 genes were related to drought tolerance. These known genes were utilized as seed nodes in the MultiPlex biological network to predict potential genes associated with drought stress.

### 4.2. Random Walk with Restart on MultiPlex Networks

Random Walk with Restart (RWR) [11] is a ranking algorithm designed to iteratively explore the overall structure of a network, revealing the degree of association between nodes. In RWR, $v_{0}$ represents the seed node, and an imaginary particle initiates random walks starting from node $v_{0}$. At each step, the particle faces two options: either jumping back to the seed node with a fixed restart probability $\left(r\right)$, or randomly walking to an adjacent node with a probability of $\left(1-r\right)$. In a MultiPlex network, particles can traverse between different nodes in the same layer or transition to different layers from the same node. After convergence, a score vector is obtained, reflecting the association between nodes and the seed node, which can be utilized to unveil critical structures in the network.

### 4.3. Multiple Probability Restart Random Walk on MultiPlex Networks

The uniform restart probability in traditional approaches limits the random walking capacity of particles in the network, often overlooking nodes with lower importance and consequently neglecting global relationships. To address this limitation, our study assesses node importance in the network using eigenvector centrality. Subsequently, we assign different restart probabilities to nodes based on their importance, with higher importance corresponding to a greater probability of restarting for walking particles. This methodology enables a thorough investigation of the extent of correlation between all nodes and seed nodes from a global perspective.

Eigenvector centrality is determined by the centrality of neighboring nodes, where the centrality of node *i* is directly proportional to the sum of the centralities of its adjacent nodes *j*. The formula for calculating the eigenvector centrality of node *i* is expressed as follows:
(1)$$E_{c}\left(i\right)=\frac{1}{\lambda }\sum_{j}A_{ij}E_{c}\left(j\right)$$
where $E_{c}\left(i\right)$ is the eigenvector value of node *i*. λ is a constant maximum eigenvalue of matrix *A*. *A* is the adjacency matrix of the graph, and only if *i* is connected to *j*, $A_{ij}=1$; otherwise $A_{ij}=0$. And $E_{c}\left(j\right)$ is the eigenvector value of node *j*. When calculating the eigenvector centrality of a node, the first step involves capturing the connections between nodes by constructing the network’s adjacency matrix. Next, initialize a vector for eigenvector centrality. Through iterative updates, the eigenvector centrality is continuously adjusted at each step by considering the eigenvector centrality of neighboring nodes until it converges to a stable vector.

In a single-layer network, the restart probability of the particle at each node can be represented by $r_{vi}=E_{c}\left(i\right)$, where $E_{c}\left(i\right)$ is the eigenvector centrality of node *i*. Therefore, the restart probability can be represented as $\bar{r}=\left[r_{v_{0}},r_{v_{1}},\ldots ,r_{v_{n}}\right]$ for each node in the single-layer network. Correspondingly, the probability of the particle’s movement at various nodes can be represented by $\bar{e\_r}=\left[1-r_{v_{0}},1-r_{v_{1}},\ldots ,1-r_{v_{n}}\right]$.

In a MultiPlex network that has L layers, the particle’s restart probability can be represented as $\bar{R}=\left[{\bar{r}}^{\left[1\right]},\ldots ,{\bar{r}}^{\left[x\right]},\ldots ,{\bar{r}}^{\left[L\right]}\right]$, where ${\bar{r}}^{\left[x\right]}$ depends on the eigenvector centrality of each layer in the MultiPlex network. When ${\bar{r}}^{\left[x\right]}$ is an n∗1 vector, $\bar{R}$ is an $\left(Ln\right)*1$ vector. The corresponding probability of the particle’s movement in the network can be represented as $\bar{E\_R}=\left[{\bar{e\_r}}^{\left[1\right]},\ldots ,{\bar{e\_r}}^{\left[x\right]},\ldots ,{\bar{e\_r}}^{\left[L\right]}\right]$.

Therefore, the distribution of the particle’s movement in the MultiPlex network from time *t* to t+1 can be represented as: (2)$${\bar{P}}_{t+1}^{T}={\left(\left(diag\left({\bar{E\_R}}^{T}\right)\right)N\right)}^{T}{\bar{P}}_{t}^{T}+z{\bar{P}}_{RS}^{T}$$

Here, *N* is the normalized adjacency matrix, and ${\bar{P}}_{RS}^{T}$ is the initial probability of the random-walking particles in the multi-layer network. The seed node is assigned a value of 1 in ${\bar{P}}_{RS}^{T}$, while the others are assigned a value of 0, which can be expressed as ${\bar{P}}_{RS}^{T}=\tau \left[{\bar{P_{0}}}^{\left[1\right]},\ldots ,{\bar{P_{0}}}^{\left[x\right]},\ldots ,{\bar{P_{0}}}^{\left[L\right]}\right]$. Here, τ represents the importance of each layer in the multi-layer network; by changing τ, certain layers can be emphasized. However, in this study, all networks are considered equally important, so $\tau =\left[1/L,\ldots ,1/L\right]$. *z* represents the probability distribution of restarting at the starting node for all nodes in the current network, determined by the current distribution of all nodes and their restarting probabilities, which can be expressed as $z=\bar{R}*{\bar{P}}_{t}^{T}$.

The detailed implementation process of RWMRP is shown in Algorithm 1.
**Algorithm 1:** Random Walk with Multi-Restart probability on MultiPlex Network**Input:** Restart probability vector $\bar{R}$, Random walk probability $\bar{E\_R}$, Transition Matrix *N*, Initial probability vector ${\bar{P}}_{RS}^{T}$, iterations iter.**Output:** A vector that displays the degree of association between all nodes in the network and the starting node ${\bar{P}}_{t+1}^{T}$1: iter=202: t=13: $z=\bar{R}*{\bar{P}}_{RS}^{T}$;4: whilet⩽iter:5: ${\bar{P}}_{t+1}^{T}={\left(\left(diag\left({\bar{E\_R}}^{T}\right)\right)N\right)}^{T}{\bar{P}}_{t}^{T}+z{\bar{P}}_{RS}^{T}$6: $z=\bar{R}*{\bar{P}}_{t}^{T}$7: t=t+18: $return\;{\bar{P}}_{t+1}^{T}$

### 4.4. Leave-One-Out Cross-Validation Strategy

The study systematically evaluated the predictive performance of the Restarted Random Walk algorithm across diverse networks using a leave-one-out cross-validation (LOOCV) strategy [9]. We established a test set comprised of genes from the common gene set between the obtained known abiotic stress genes and genes present in the MultiPlex biological network. The test set genes were successively singled out and treated as left-out genes, while the remaining genes functioned as seed nodes for the Restarted Random Walk algorithm. This algorithm computed the proximity between all nodes in the network and the seed nodes, generating scores and rankings for each node. Following each iteration of the Restarted Random Walk, the scores and rankings for the left-out genes were documented. This process iterated until all test set genes were systematically tested, and their rankings were sequentially recorded. The detailed implementation of the leave-one-out strategy is outlined in Algorithm 2 [49].
**Algorithm 2:** Leave-Out Cross-Validation Strategy**Input:** Abiotic stress-related gene set: test_set.**Output:** Ranking set for left-out genes: left_out_gene_result1: **function** LOOCV_S $\left(test\_set\right)$2: left_out_gene_result=Null3: **for** i=1 **to** length $\left(test\_set\right)$ **do**4: left_out_gene_result = $test\_set\left[i\right]$5: $test\_genes=test\_set\;\left[1,2,\ldots ,i-1,i+1,\ldots ,n\right]$6: $all\_genes\_rank=RWMRP\;\left(test\_genes\right)$7: left_out_generank8: $left\_out\_gene\_result.append\;\left(left\_out\_generank\right)$9: **end for**10: **return** left_out_gene_result11: **end function**

### 4.5. Association Analysis

Correlated genes may have similar functions [2]. Hence, genes exhibiting interactions with multiple known drought stress genes are more probable to be associated with drought stress. For each potential gene, denoted as $PG_{j}\;\left(j=1,2,\ldots ,n\right)$, interaction scores with $KG_{i}\;\left(i=1,2,\ldots ,m\right)$ were acquired from the STRING database, designated as $S=\;\left(PG_{j},KG_{i}\right)$ [9]. $KG_{i}$ represents known drought stress genes obtained from the China Rice Data Center. Any potential gene with 3 or more $S=\left(PG_{j},KG_{i}\right)\geq 0.9$ was considered a candidate gene for drought stress.

### 4.6. qRT-PCR Analysis

The qRT-PCR method can be used to analyze the expression levels of candidate genes under drought stress, thereby validating whether these genes are involved in drought response. NPB seeds were cultivated hydroponically until the three-leaf stage, followed by the simulation of drought conditions using a 15% PEG6000 solution [50]. Samples of rice leaves were collected at 0 h, 4 h, and 24 h after drought treatment, with three biological replicates for each time point. Primers for the candidate genes were designed using the Primer-BLAST tool from NCBI (https://www.ncbi.nlm.nih.gov/tools/primer-blast/ (accessed on 1 July 2024)). The rice ACTIN2 gene served as the internal reference gene. Calculation of the expression levels of the candidate genes was performed using the $\left(2^{-\Delta \Delta Ct}\right)$ method.

## 5. Conclusions

In this study, we established a MultiPlex biological molecular network for rice and introduced the RWMRP algorithm, which assigns distinct restart probabilities to each node in the network based on eigenvector centrality. We systematically compared the gene prediction performance of RWMRP and RWR across single-layer, dual-layer, and MultiPlex networks. The experimental results consistently demonstrate that MultiPlex networks significantly enhance the algorithm’s predictive capabilities. In the MultiPlex networks, RWMRP outperforms RWR.

We applied the RWMRP algorithm to a multi-biomolecular network in rice to predict genes associated with drought stress. Through association analysis, we further identified candidate genes that are highly associated with known drought-responsive genes. We performed qRT-PCR analysis on these candidate genes, ultimately identifying 10 drought stress-responsive genes in rice. This research contributes valuable insights to the understanding of regulatory mechanisms governing abiotic stress in rice and holds reference value for future studies in this field.

## Figures and Tables

**Figure 1 ijms-25-09216-f001:**
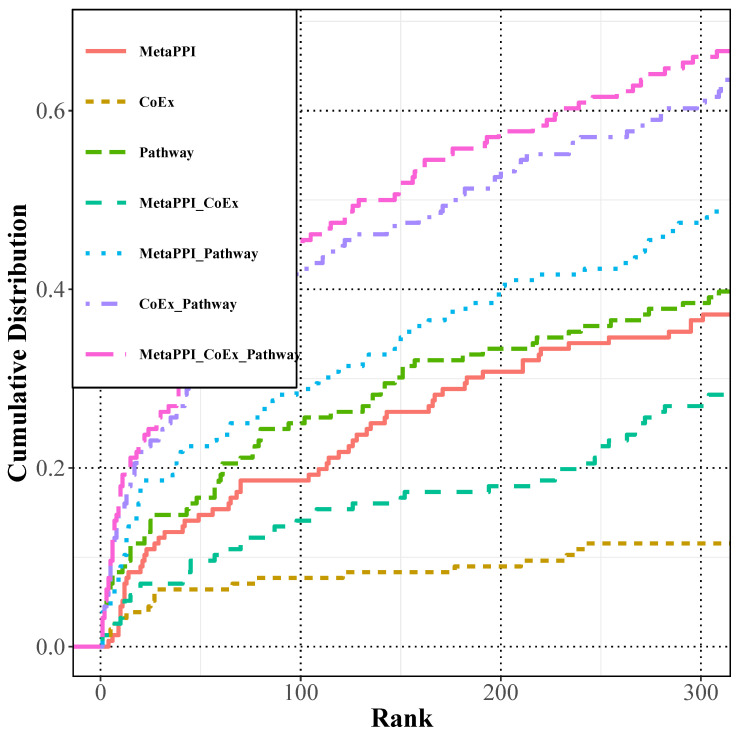
The prediction performance of RWR on various biological molecular networks. The cumulative distribution function represents the ranking of omitted genes in the LOOCV strategy across different biological molecular networks. The X-axis represents the ranking of nodes in the network, and the Y-axis represents the ratio of the number of omitted genes before the current ranking to the total number of genes in the test set. Among them, RWR consistently outperformed other networks on the multi-layered biological molecular network MetaPPI_CoEx_Pathway network.

**Figure 2 ijms-25-09216-f002:**
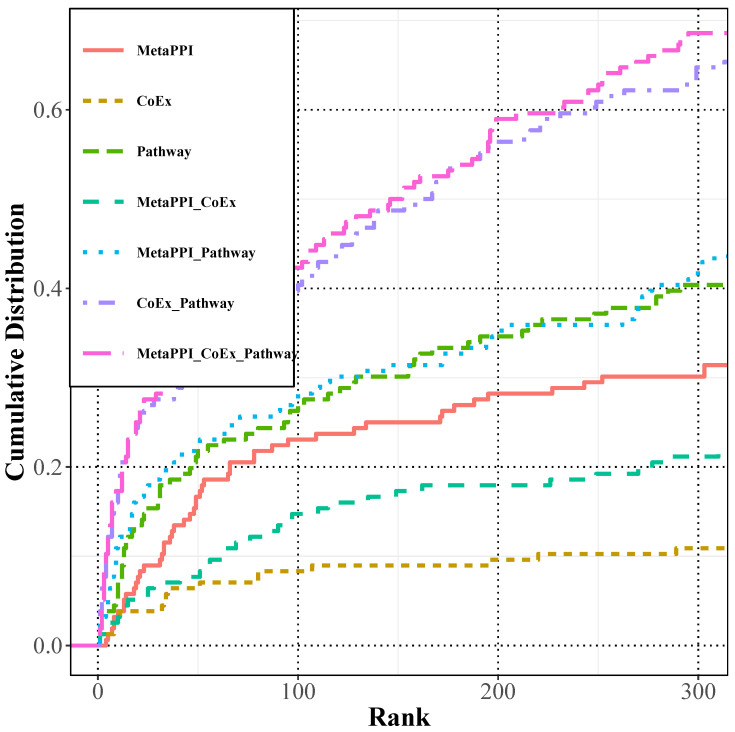
The prediction performance of RWMRP on various biological molecular networks. Similarly, RWMRP consistently demonstrates superior performance over other networks on the multi-biological molecule network MetaPPI_CoEx_Pathway.

**Figure 3 ijms-25-09216-f003:**
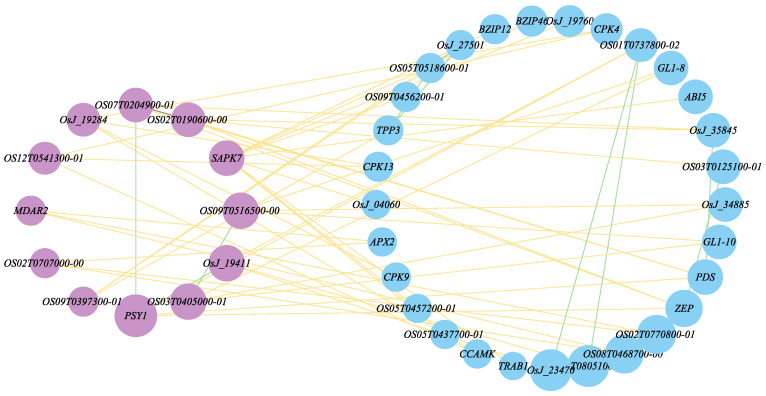
Association analysis of drought stress potential genes with known drought stress genes. The figure retains the selected candidate genes as well as known drought-resistant genes. Nodes in light blue on the right signify known drought stress genes, while nodes in light purple on the left represent candidate genes. Each yellow edge indicates an interaction between a known gene and a candidate gene.

**Figure 4 ijms-25-09216-f004:**
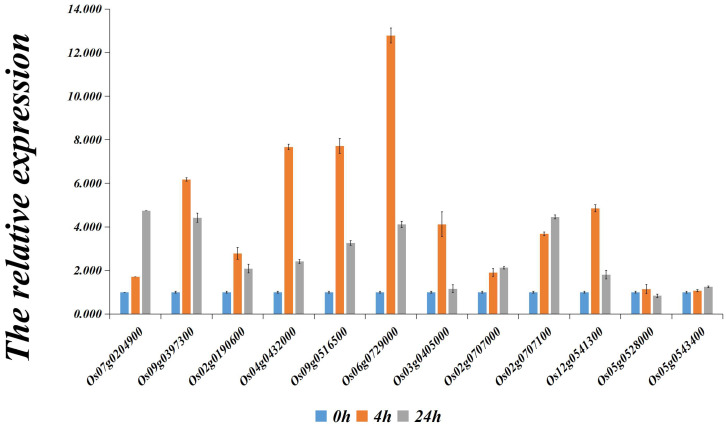
The results of gene expression changes of drought candidate genes after treatment with 15% PEG6000 reagent for 0 h, 4 h, and 24 h.

**Figure 5 ijms-25-09216-f005:**
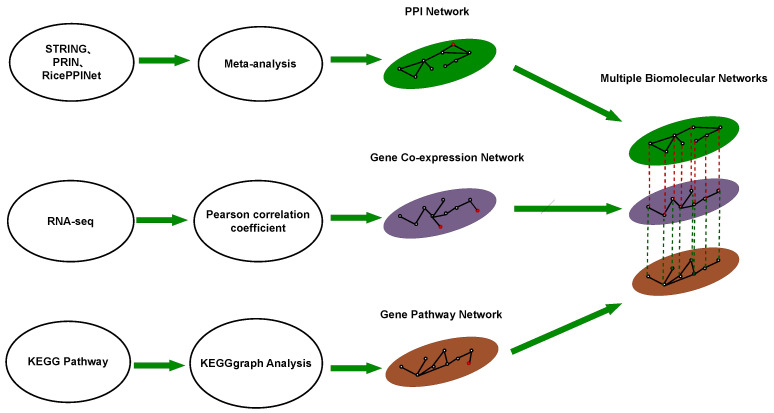
The construction process of MultiPlex biological networks. During the process of network fusion, we retained the common nodes on the PPI network, gene co-expression network, and gene pathway network. The red nodes represent the deleted nodes.

**Table 1 ijms-25-09216-t001:** The results of enrichment analysis of potential genes for drought stress.

GO Term	Number of Genes	Ontology	Description	*p*-Value
GO:0008152	213	BP	Metabolic process	4.85 × $10^{-82}$
GO:0009605	14	BP	Response to environmental stimulus	2.22 × $10^{-3}$
GO:0009266	7	BP	Response to temperature stimulus	3.86 × $10^{-4}$
GO:0006950	39	BP	Response to abiotic stress	1.19 × $10^{-9}$
GO:0009737	7	BP	Response to abscisic acid	3.75 × $10^{-5}$
GO:0004674	16	MF	Protein serine/threonine kinase activity	8.26 × $10^{-4}$
GO:0046905	2	MF	15-cis-phytoene synthase activity	1.28 × $10^{-4}$
GO:0003825	2	MF	Alpha, alpha-trehalose-phosphate synthase (UDP-forming) activity	1.17 × $10^{-3}$
GO:0016688	2	MF	L-ascorbate peroxidase activity	1.49 × $10^{-3}$
GO:0005737	139	CC	Cytoplasm	5.32 × $10^{-42}$
GO:0043229	124	CC	Intracellular organelle	6.66 × $10^{-19}$
GO:0043231	120	CC	Intracellular membrane-bounded	2.66 × $10^{-20}$

**Table 2 ijms-25-09216-t002:** Annotation of candidate genes related to drought stress.

Candidate Genes	Name in STRING	Annotation
Os07g0204900	OS07T0204900-01	Zeta-carotene desaturase; Catalyzes the conversion of zeta-carotene to lycopene via the intermediary of neurosporene. It carries out two consecutive desaturations (introduction of double bonds) at positions C-7 and C-7′
Os09g0397300	OS09T0397300-01	Os09g0397300 protein; Putative trehalose-6-phosphate synthase/phosphatase
Os02g0190600	OS02T0190600-00	Putative lycopene beta-cyclase
Os04g0432000	SAPK7; OsJ_14855	Serine/threonine-protein kinase SAPK7
Os09g0516500	OS09T0516500-00	NAD dependent epimerase/dehydratase family protein
Os06g0729000	PSY1	cDNA clone: J023057D05, full insert sequence; Phytoene synthase-like
Os03g0405000	OS03T0405000-01	Reticulon-like protein; NAD dependent epimerase/dehydratase family protein
Os02g0707000	OS02T0707000-00	Monodehydroascorbate reductase, putative, expressed
Os02g0707100	MDAR2	Monodehydroascorbate reductase 2, peroxisomal; Catalyzes the conversion of monodehydroascorbate to ascorbate, oxidizing NADH in the process. Ascorbate is a major antioxidant against reactive oxygen species (ROS) and nitric oxide (NO)
Os12g0541300	OS12T0541300-01	Respiratory burst oxidase, putative, expressed
Os05g0528000	OsJ_19284	Putative cytochrome b245 beta chain
Os05g0543400	OsJ_19411	cDNA clone: J023072A14, full insert sequence; Os05g0543400 protein; Putative farnesyl pyrophosphate synthase; cDNA clone: J023072A14, full insert sequence; cDNA clone: J033069I20, full insert sequence; Belongs to the FPP/GGPP synthase family

## Data Availability

Data is contained within the article and Appendix A.

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
