# Peer review of "Identification of Drought Stress-Responsive Genes in Rice by Random Walk with Multi-Restart Probability on Multi-Plex Biological Networks"

_ijms, 2024, doi:10.3390/ijms25179216_

Round 1
Reviewer 1 Report (Previous Reviewer 2)
Comments and Suggestions for Authors
With the inclusion of the qRT-PCR experiments the model has been validated. paper is now ready for acceptance.
Author Response
非常感谢您对我们稿件质量和研究内容的肯定。祝你生活幸福!
Reviewer 2 Report (New Reviewer)
Comments and Suggestions for Authors
This article describes the identification of drought stress-responsive genes in rice by a random walk with multi-restart probability on multiplex biological networks.
To improve the manuscript, the authors should take the following considerations:
(1) The authors presented the construction process of multiplex biological networks in Figure 1. The authors the common nodes on the protein-protein interaction (PPI) network, gene co-expression network, and gene pathway network during the process of network fusion. The authors should provide and discuss the PPI network by the density functional theory (DFT) modeling.
(2) The authors presented the association analysis of drought stress potential genes with known drought stress genes in Figure 4. The authors should discuss in detail the molecular mechanisms involved in drought stress in rice for the coherency of the manuscript.
(3) The authors presented the results of gene expression changes of drought candidate genes after treatment with 15% PEG6000 reagent for 0 h, 4 h, and 24 h in Figure 5. It is noted that the authors should provide the original gene expression in the Y-axis in an additional figure for readability and reproducibility purposes.
The submitted manuscript has significant scientific insights, and the conclusions are soundly supported by the protein interaction networks, gene co-expression networks, and gene pathway networks based on rice multi-omics data, with each node corresponding to genes or proteins. However, the manuscript requires major revisions before being accepted in the Special Issue: Advance in Plant Abiotic Stress in the well-circulated International Journal of Molecular Sciences.
Comments on the Quality of English LanguageAbstract: Exploring drought stress-responsive genes in rice is essential for breeding drought-resistant varieties. Rice drought resistance is controlled by multiple genes, and mining drought stress-responsive genes solely based on single omics data lacks stability and accuracy. Multi-omics correlation analysis and biological molecular network analysis provide robust solutions. This study proposed a random walk with a multi-restart probability (RWMRP) algorithm, based on the restarted random walk (RWR) algorithm, to operate on rice multiplex biological networks. It explores the interactions between biological molecules across various levels and ranks potential genes. RWMRP uses eigenvector centrality to evaluate node importance in the network and adjusts restart probabilities accordingly, diverging from the uniform restart probability employed in RWR. In the random walk process, it can be better to consider the global relationships in the network. Firstly, we constructed a multiplex biological network by integrating the rice protein-protein interaction, gene pathway, and gene co-expression networks. Then, we employed RWMRP to predict potential genes associated with rice tolerance to drought stress. Enrichment and correlation analyses resulted in the identification of 12 drought-related genes. We further conducted quantitative real-time polymerase chain reaction (qRT-PCR) analysis on these 12 genes, ultimately identifying 10 genes responsive to drought stress.
Round 2
Reviewer 2 Report (New Reviewer)
Comments and Suggestions for Authors
Dear Authors: Many thanks for your sincere efforts in improving your manuscript. The revised article is highly satisfactory and merits acceptance for publication in the Special Issue: Advance in Plant Abiotic Stress in the International Journal of Molecular Sciences.
This manuscript is a resubmission of an earlier submission. The following is a list of the peer review reports and author responses from that submission.
Round 1
Reviewer 1 Report
Comments and Suggestions for Authors
Dear Authors,
Reviewer comments ijms-3051091
The manuscript entitled „Identification of drought stress-responsive genes in rice by random walk with multi-restart probability on multiplex biological networks“ represents a bioinformatics study using random walk with multi-restart probability algorithm (RWMRP) approach to identify candidate genes for drought tolerance in rice. I can recommend the manuscript for publication in IJMS.
However, I have some comments on the present manuscript:
1/ In Materials and methods, part 2.1. Data sets and biological molecular network, the accession number and date of access have to be added to every data set employed in the analysis.
2/ In Discussion, I think that potential biological functions of the drought-responsive candidate genes in rice identified in the present study should be discussed and a model providing a complex view on the roles of these candidate genes in rice response to drought should be added as a new figure.
3/ Formal comments on the text related to English language and style:
Introduction, line 64: Join the two statement into one as follows: „These algorithms have innovatively approached network construction enhancing their performance in the random walk processes.“
Introduction, lines 77-78: The statement is very long and not much comprehensive; I think that it might be modified as follows: „The predicted genes for drought stress underwent further screening through enrichment analysis and drought stress candidate genes through gene association analysis resulting in the identification of rice drought stress-responsive genes.“
Materials and methods, line 88: Add a space between two statements, i.e., „respectively“ and „To ensure consistency in gene nomenclature…“
Final recommednation: Accept after a minor revision.
Comments on the Quality of English Language
Dear Authors,
Reviewer comments ijms-3051091
The manuscript entitled „Identification of drought stress-responsive genes in rice by random walk with multi-restart probability on multiplex biological networks“ represents a bioinformatics study using random walk with multi-restart probability algorithm (RWMRP) approach to identify candidate genes for drought tolerance in rice. I can recommend the manuscript for publication in IJMS.
However, I have some comments on the present manuscript:
1/ In Materials and methods, part 2.1. Data sets and biological molecular network, the accession number and date of access have to be added to every data set employed in the analysis.
2/ In Discussion, I think that potential biological functions of the drought-responsive candidate genes in rice identified in the present study should be discussed and a model providing a complex view on the roles of these candidate genes in rice response to drought should be added as a new figure.
3/ Formal comments on the text related to English language and style:
Introduction, line 64: Join the two statement into one as follows: „These algorithms have innovatively approached network construction enhancing their performance in the random walk processes.“
Introduction, lines 77-78: The statement is very long and not much comprehensive; I think that it might be modified as follows: „The predicted genes for drought stress underwent further screening through enrichment analysis and drought stress candidate genes through gene association analysis resulting in the identification of rice drought stress-responsive genes.“
Materials and methods, line 88: Add a space between two statements, i.e., „respectively“ and „To ensure consistency in gene nomenclature…“
Final recommednation: Accept after a minor revision.
Reviewer 2 Report
Comments and Suggestions for Authors
The paper is very original as describes an algorithm to identify stress related genes by integrating information from different sources, as an outcome the authors obtain a list of genes already related to stress in rice, and a list of probable candidate genes hypotetically related to the stress response.
The major problem of this paper is that there is no wet experiment to confirm the validity of the new results. What is good we already knew, but we have the doubt of whether the proposed new genes are indeed stress related genes, useful to develop biotechnological applications, or merely false positives of the algorithm. So, authors must somehow confirm the validity of the candidate genes to stregthen te conclusions of this paper, that in the present version are merely speculative. Authors should:
a) Perform qRT analysys on the candidate genes in figure 4 to check that they are stress regulated.
b) Look for orthologues of the candidate genes in figure 4 in other organisms and check whether they have been related to stress response.
Round 2
Reviewer 2 Report
Comments and Suggestions for Authors
I really disagree with the authors about the difficulty of doing qRT PCR analysis, as this is a standard and routine technique. With out this crucial confirmation, the validity of the results is deeply compromised, so I cannot recommend publication.